# Polymorphism in S(+)Clopidogrel-Picrate: Insights from X-ray Diffraction, Vibrational Spectroscopy, Thermal Analysis, and Quantum Chemistry

**Aleksandar Cvetkovski** [1,*] **, Petre Makreski** [2] **, Ljupcho Pejov** [2,3,4] **, Monika Stojanovska Pecova** [5] **, Valerio Bertolasi** [6] **, Paola Gilli** [6] **and Leonard R. MacGillivray** [7]

1 Faculty of Medical Sciences, Goce Delcev University, No. 10-A, P.O. Box 201, 2000 Stip, North Macedonia
2 Institute of Chemistry, Faculty of Natural Sciences and Mathematics, Ss. Cyril and Methodius University in Skopje, 1000 Skopje, North Macedonia; petremak@pmf.ukim.mk (P.M.); ljupcop@pmf.ukim.mk (L.P.)
3 Department of Chemistry, Bioscience and Environmental Engineering, Faculty of Science and Technology, University of Stavanger, 4021 Stavanger, Norway
4 The Polytechnic School, Ira A. Fulton Schools of Engineering, Arizona State University, Mesa, AZ 85004, USA
5 Research and Development, Alkaloid AD, Blvd. Aleksandar Makedonski 12, 1000 Skopje, North Macedonia; mstojanovska1@alkaloid.com.mk
6 Dipartimento di Scienze Chimiche e Farmaceutiche e Centro di Strutturistica Diffrattometrica, Università di Ferrara, Via Borsari 46, 44121 Ferrara, Italy; m38@unife.it (V.B.); paola.gilli@unife.it (P.G.)
7 Department of Chemistry, The University of Iowa, E331 Chemistry Building, Iowa City, IA 52242, USA; len-macgillivray@uiowa.edu
* Correspondence: aleksandar.cvetkovski@ugd.edu.mk

**Abstract:** The crystal structures of two pseudopolymorphic forms of S(+)clopidogrel–picrate are reported. Form **1** crystallizes in the monoclinic space group $P2_1$ with an ionic couple S(+)ClopH+·Pic$^−$ and a molecule of solvent ethanol in the asymmetric unit, while Form **2** crystallizes in the monoclinic space group $C2$ with two ionic couples in the asymmetric unit. The configurations and conformations of the ionic couples, held together by ionized +N-H···O hydrogen bonds, are nearly identical in the structures. The self-assembly properties are compared with reported clopidogrel salts, including those used in pharmaceutical formulations. The hydrogen bonds are discussed in reference to the general corresponding behavior of the N-bases picrates and the properties of the acid-base coformers. The preparations of the pseudopolymorphs were optimized toward two different methods: solvent evaporation and mechanochemical treatment. Reproducibility to generate the single crystalline phases was confirmed by thermal and vibrational spectroscopic properties. Periodic third-order density-functional tight binding (DFTB3) calculations predict rather small energy difference between the two pure phases of polymorphs **1** and **2**. However, the included solvent molecules in Form **1** decrease the lattice energy for ~10.5 kcal mol$^{-1}$, which leads to a lower $\Delta E_{\text{latt.}}$ lattice energy in comparison to Form **2** (by ~7.3 kcal mol$^{-1}$). All predicted trends are in line with the experimentally observed formation of Form **1** instead of its simulated non-solvated Form **1**.

**Keywords:** clopidogrel; Polymorphism; co-crystals; crystal structure; solid-state properties

## 1. Introduction

Clopidogrel hydrogen sulfate (ClopH$^+$·HSO$_4$$^−$), or Plavix$^®$ (BMS-Sanofi), is a potent antiplatelet drug that acts as a selective and irreversible inhibitor of ADP-induced platelet aggregation. The salt is a thienopyridine class inhibitor of the P2Y12 ADP platelet receptors located on the membranes of platelet cells. From a molecular perspective, the salt contains the single enantiomer of (S)-(+)-methyl-2-(2-chlorophenyl)-2-(6,7-dihydro-4*H*-thieno[3,2-*c*]pyridin-5-yl)acetate along with the hydrogen sulfate ion. The free base of clopidogrel (Figure 1) is an oily liquid.

**Figure 1.** Molecular structure of clopidogrel free base.

The action of clopidogrel involves behavior as a prodrug that is capable of inhibiting the ADP P2Y12 receptor, a key mediator of platelet aggregation. Clopidogrel is activated in the liver by cytochrome P450 enzymes, particularly CYP2C19, which is an important drug-degrading enzyme that catalyzes the biotransformation of many chemicals, including antidepressants, barbiturates, proton pump inhibitors, and antimalarials and antitumor drugs. Clopidogrel is used to prevent myocardial infraction and stroke in people at high risk and in stent-associated thrombosis [1–3].

Clopidogrel exists in two enantiomeric forms, the R(−) and S(+) isomers, with the dextrorotatory isomer exhibiting pharmacological activity [4]. Different polymorphic forms of clopidogrel–hydrogen sulfate are known, with Forms I and II being used in pharmaceutical formulations [5]. The crystal structure of Form I crystallizes monoclinic [6], while Form II crystallizes orthorhombic [7] (CSD reference codes FUQMOU01 and FUQMOU). Two crystal structures of the cytochrome P450 2B4 active site mutant F297A in a complex with clopidogrel have also been reported [8,9] (PDB ID 4H1N and 3ME6). To date, only one other structure of the S(+)clopidogrel salt is known: S(+)clopidogrel isopropyl sulfate (reference code YEXHOZ) [10]. As an underlying motivation to obtain new salts of this important drug, a systematic co-crystallization screening for molecular salts of clopidogrel with strong organic acids was performed. Co-crystallization experiments with drugs, nutraceuticals, or excipients as co-formers did not result in crystalline phases of sufficient quality until we succeeded in obtaining single crystals of two pseudopolymorphic salts with picric acid (Pic) ($pK_a = 0.36$), the structures of which are reported here. Picric acid is a strong organic acid traditionally used in the crystallization of basic compounds, with several hundreds of mixed crystals containing the picrate anion in the CSD [11].

The crystal packing and the relationships between $\Delta pK_a$ and the H-bond strength of a large series of H-bonded adducts formed by picric acid with nitrogen bases have recently been investigated [12,13], according to which the H-bond formed by clopidogrel with picric acid can be expected to be of medium–strong strength. According to the p$K_a$ *equalization principle* [14–17], which states that the strength of the D–H⋯: A bond increases as $\Delta pK_a$ decreases and that this strength reaches a maximum as $\Delta pK_a$ approaches zero, this p$K_a$ difference also determines the geometry and energetics of the charged $^+$N–H⋯O$^-$ bonds linking cations and anions within the ionic couples.

In our work, two different techniques of crystalline phase preparation were used: conventional solvent evaporation versus eco-friendly mechanochemical synthesis. Scaling of both approaches confirmed the success and reproducibility to obtain the two pseudopolymorphs, each with distinctive thermodynamic and vibrational spectroscopic features.

## 2. Materials and Methods

### 2.1. Syntheses of Single Crystals

All reagents and solvents used were purchased from Sigma Aldrich (Burlington, MA, USA), except for S(+)clopidogrel–hydrogen sulfate, which was from Aarti Drugs Ltd. (ADL) (Mumbai, India). Form 1 was prepared by dissolving clopidogrel–hydrogen sulfate and

picric acid in a 1:2 molar ratio in 98% ethanol. The solution was slowly evaporated until yellow crystals precipitated. Form 2 was prepared by dissolving the free base of clopidogrel and picric acid in a 1:1 molar ratio in a methanol/*n*-pentanol mixture (50% *v/v*). The clear solution slowly evaporated until precipitation of yellow crystals. The clopidogrel free base was prepared by treating clopidogrel–hydrogen sulfate with sodium bicarbonate and then extracting the neutral base with dichloromethane, resulting in a viscous oily liquid. The purity of the S(+) isomer was confirmed by optical rotation measurement, $[\alpha]_D^{23} = +47$ (c = 1.06 in methanol) [18].

For $(C_{16}H_{17}ClNO_2S)^+ \cdot (C_6H_2N_3O_7)^- \cdot C_2H_6O$: Anal. calcd (%): C, 48.24; H, 4.18; N, 9.38; S, 5.36; Cl, 5.93. Found, (%): C, 48.26; H, 4.20; N, 9.40; S, 5.38; Cl, 5.95.

For $(C_{16}H_{17}ClNO_2S)^+ \cdot (C_6H_2N_3O_7)^-$: Anal. calcd, (%): C, 47.92; H, 3.45; N, 10.16; S, 5.81; Cl, 6.42. Found, (%): C, 47.94; H, 3.46; N, 10.18; S, 5.84; Cl, 6.44.

## 2.2. Methods for Bulk Preparation of Crystalline Forms 1 and 2

### 2.2.1. Solvent Evaporation

A viscous oily liquid of clopidogrel free base was isolated via dichloromethane extraction from commercial clopidogrel, S(+)clopidogrel–hydrogen sulfate in alkaline water extraction media. Subsequently, the base and the powdered crystalline picric acid, in a stoichiometric 1:1 ratio, was dissolved in 98% ethanol and *n*-pentanol at ambient temperature, respectively. Each of the two alcohol solutions was left at room temperature for slow evaporation of the solvent until crystalline solid phases were formed. The crystalline solid sediment grown in the ethanol batch was designated sample A (resembles Form **1**), while the solid phase that grew after evaporation of *n*-pentanol was designated sample B (resembles Form **2**). Both samples were further analyzed to evaluate thermal and vibrational spectroscopic properties.

### 2.2.2. Mechanochemical Treatment

A weighted mass of powdered picric acid (ground by hand) was added to a weighted mass of oily viscous clopidogrel free base in a stoichiometric ratio of 1:1. The sticky mixture was divided into two mortars, and the mixture in each mortar was kneaded separately for 20 min with a pestle in the presence of drops of ethanol (sample C) or *n*-pentanol (sample D). After vigorous kneading, the grinding was stopped and the mechanochemically obtained dried solids were designated as sample C and D (which resembled Form **1** and **2**, respectively). The samples were also collected and analyzed in the solid-state for further characterization.

## 2.3. Single-Crystal X-ray Determination

The crystal data of structures **1** and **2** were collected at room temperature using a Nonius KappaCCD diffractometer (Bruker, Billerica, MA, USA) with graphite monochromatic Mo-K$\alpha$ radiation. The data sets were integrated with the Denzo-SMN package (New York, NY, USA) [19] and corrected for Lorentz and polarization effects. The structures were solved by direct methods using the SIR97 [20] system of programs.

The structure of Form 1 was refined by the full-matrix least-squares method with the non-H atoms of the S(+)ClopH$^+ \cdot$Pic$^-$ salt anisotropically and hydrogens isotropically, except for those of the methyl group and those belonging to the disordered solvent molecule ethyl group, which were included on calculated positions, riding on their carrier atoms. The two *ortho*-NO$_2$ groups of the picrate anion were disordered. Split occupancies of 0.5 and 0.5 were applied to the O4 and O5 atoms, while split occupancies of 0.6 and 0.4 were applied to O8 and O9. Also, the solvent molecule of ethanol in the structure was found to be disordered and split occupancies of 0.5 and 0.5 were applied to the ethylic CH$_3$-CH$_2$ group. Crystal data of Form **1**: $(C_{16}H_{17}ClNO_2S)^+ \cdot (C_6H_2N_3O_7)^{--} \cdot C_2H_6O$, *M* = 596.99, Monoclinic, Space group *P*2$_1$ (No.4), *a* = 12.8272(3) Å, *b* = 7.5546(1) Å, *c* = 15.2917(4) Å, *β* = 113.3015(8)°, *V* = 1360.97(5) Å$^3$, *Z* = 2, $D_c$ = 1.457 g cm$^{-3}$, $\mu$(Mo-K$\alpha$) = 0.280 mm$^{-1}$, *T* = 295 K, 6221 independent reflections, $\theta \le 28.00°$, 5459 observed reflections [$I \ge 2\sigma(I)$], $R_1$ = 0.0593 (observed

reflections), $wR_2$ = 0.1717 (all reflections), GOF = 1.032, 442 parameters. Chirality: N1: (R), C8: (S), Flack parameter = −0.12(8) [21].

The structure of Form **2** was refined by full-matrix least-squares with anisotropic non-H atoms of both the S(+)ClopH$^+$·Pic$^-$ ionic couples in the asymmetric unit and hydrogens included on calculated positions, riding on their carrier atoms, except the N-H hydrogens which were refined isotropically. Crystal data of Form **2**: $(C_{16}H_{17}ClNO_2S)^+$·$(C_6H_2N_3O_7)^-$, $M$ = 550.92, Monoclinic, Space group *C*2 (No.5), $a$ = 25.4658(6) Å, $b$ = 14.6087(4) Å, $c$ = 13.9702(5) Å, $\beta$ = 108.8426(10)°, $V$ = 4918.7(3) Å$^3$, $Z$ = 8, $D_c$ = 1.488 g cm$^{-3}$, $\mu$(Mo-K$\alpha$) = 0.300 mm$^{-1}$, $T$ = 295 K, 9544 independent reflections, $\theta \leq 26.50°$, 6207 observed reflections $[I \geq 2\sigma(I)]$, $R_1$ = 0.0540 (observed reflections), $wR_2$ = 0.1190 (all reflections), GOF = 1.070, 685 parameters. Chirality N1A and N1B: (R), S1A and S1B: (S), Flack parameter −0.03(7) [21].

All calculations were performed using SHELXL-97 [22] and PARST [23] implemented in the WINGX [24] system. The complete crystal data are given in Supplementary Table S1, and the selected bond distances, bond angles, and torsion angles are shown in Supplementary Table S2.

Crystallographic data have been deposited in the Cambridge Crystallographic Data Centre and allocated the deposition numbers CCDC 1448941 and 1448942. These data can be obtained free of charge via https://www.ccdc.cam.ac.uk/structures/ (accessed on 14 December 2023) or on application to CCDC, Union Road, Cambridge, CB2 1EZ, UK [fax: +44-1223-336033, e-mail: deposit@ccdc.cam.ac.uk].

ORTEP [25] views of the S(+)ClopH$^+$·Pic$^-$ salts that define the pseudopolymorphic crystals **1** and **2** are shown in Figures **2** and **3**, respectively. A selection of bond distances, angles and torsion angles, and the hydrogen bond (H-bond) geometries are reported in Supplementary Tables S2 and S3.

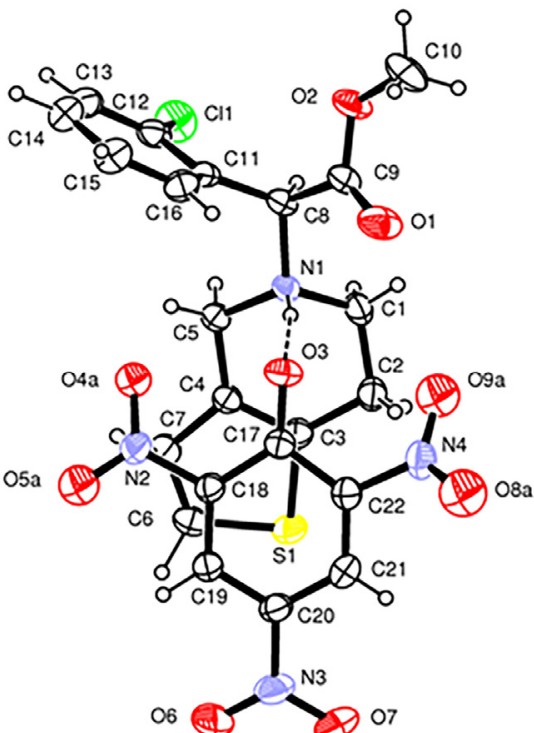

**Figure 2.** ORTEP view of the S(+)ClopH$^+$·Pic$^-$ salt of Form **1** showing the thermal ellipsoids at 30% probability level.

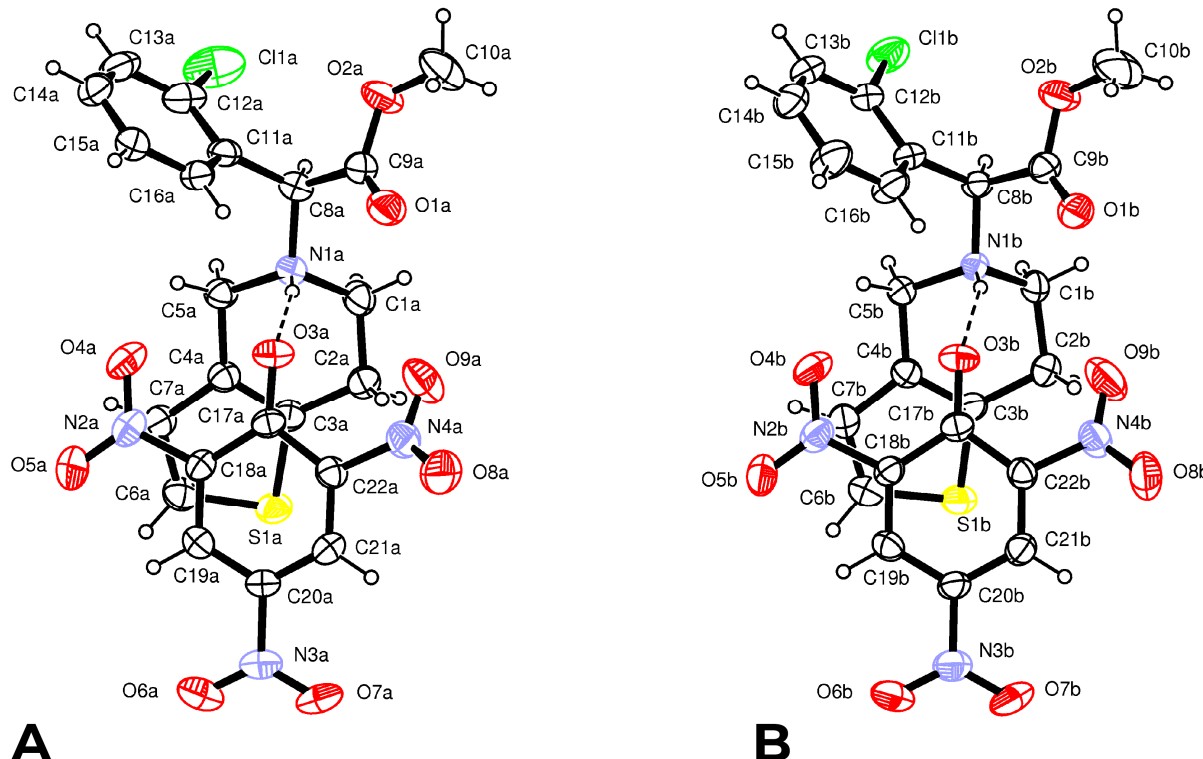

**Figure 3.** ORTEP view of the two independent ionic couples (**A**,**B**) of S(+)ClopH$^+$·Pic$^-$ salt of Form **2** showing the thermal ellipsoids at 30% probability level.

The H-bond energies, $E_{HB}$ in kcal mol$^{-1}$, of the D−H...A bonds (D, A = N, O) in Forms **1** and **2** are evaluated by the Lippincott and Schroeder (LS) method [26–28] as a function of the D...A distance and D-H...A angle. Although more sophisticated methods for estimation of the strength and nature of non-covalent intermolecular interactions have been proposed in the literature, such as the much quoted Bader "atoms-in-molecules" (AIM) electron density analysis technique, the newer NCI technique—which is essentially based on analysis of the reduced density gradient—and the "natural bond orbital" concept (NBO) devised by Weinhold [29–31], relying on the data obtained by the LS analysis is justified by several reasons. All quantitative conclusions based on AIM or NCI approaches are based on the correlation of the data computed for a particular electron density. Due to the size of the presently studied system, the DFTB technique is suitable to compute the energetic properties in the present study. More detailed analyses of electron density-related properties will be the subject of our subsequent investigations, using more exact and advanced periodic DFT methods.

### 2.4. X-ray Powder Diffraction

The reproducibility of the two preparation procedures used (single crystals vs. bulk powder) was optimized by comparing the calculated diffractograms of Forms **1** and **2** with the collected X-ray powder diffraction (XRPD) patterns of the samples A–D. X-ray powder diffraction measurements were carried out using a Rigaku Ultima IV diffractometer equipped with Cu*Kα* radiation (λ = 1.54178 Å) from an X-ray tube at 40 kV and a current of 40 mA. The *Kβ* filter was used, and the following optics were employed: divergence slit 2/3 deg, divergence height slit 10 mm, and scattering slit 8 mm. Diffraction data were collected over a 2*θ* range from 4 to 50 degrees at a constant rate of 4 degrees/min using a high-speed position-sensitive linear (1D) D/teX Ultra detector.

## 2.5. Thermal Analysis

Differential scanning calorimetry (DSC) measurements were performed on a Netzsch DSC 204 F1 Phoenix instrument (Burlington, MA, US), in aluminum pans with perforated lids, with a temperature range from room temperature to 170 °C. Thermogravimetric (TG)/DTG analyses were performed on a Netzsch TG 209 F1 Iris thermogravimetric analyzer, in $Al_2O_3$ pans, with a temperature range from room temperature to 400 °C. All measurements were carried out at a heating rate of 10 K/min under a dynamic nitrogen atmosphere (30 mL/min).

## 2.6. Vibrational Spectroscopic Analysis

The Varian-660 Fourier transform infrared (FT IR) spectrometer (Agilent, Santa Clara, CA, USA) was used to acquire the FT IR spectra. Attenuated total reflectance (ATR) spectra (resolution 4 $cm^{-1}$, 16 scans per spectrum) were collected using the GladiATR module with diamond crystal (PIKE technologies, Madison, WI, USA) in the 4000–400 $cm^{-1}$ range. Raman spectra at room temperature (20 °C) were recorded using the Horiba JobinYvon LabRam 300 Infinity micro-Raman multichannel spectrometer (Piscataway, NJ, USA). An Olympus MPlanN confocal microscope with a ×50 objective (long distance) (Tokyo, Japan) was chosen for magnification. To focus the laser beam, a confocal hole of about 2 μm was used and the position on the sample surface was adjusted with a motorized x-y stage. The Raman effect was obtained using the 632.8 nm line of a He:Ne laser with a power of 1.9 mW. The backscattered radiation (180° configuration) was analyzed with a 1800 lines/mm grating monochromator. Raman intensities were collected with a thermo-electrically cooled CCD array detector. The resolution of the system ("apparatus function") was 3 $cm^{-1}$ and the wavenumber accuracy $\pm 1$ $cm^{-1}$ (both calibrated with the Rayleigh line and the 520.5 $cm^{-1}$ line of a Si standard).

## 2.7. Computational Details

Our theoretical approach in the present study is based on the density functional tight binding approach (DFTB) [32–34]. We hereby rely on the so-called third-order DFTB (DFTB3) [35]. More technical details about the methodology and the parametrization are provided in the Supplementary Material (Theoretical part).

All DFTB computations were carried out with the DFTB+ code [35,36], using the 3ob Slater–Koster parameter set [37–39] and the corresponding Hubbard parameter set; the damping exponent $\zeta$ value in (3) was set to 4.0. Geometry optimizations of atomic positions within the unit cell of the studied 3D periodic systems were carried out, employing the rational function-based optimization algorithm. Careful testing of the choice of the Monkhorst–Pack grid [40] for *k*-point sampling has been carried out for all studied systems, controlling the convergence of energy (which was better than $10^{-4}$ eV for all systems) as well as geometry. Finally, productive calculations have been performed with the Γ—point centered $8 \times 8 \times 8$ Monkhorst-Pack grid.

## 3. Results and Discussion

Form **1** crystallizes in the $P2_1$ space group with one protonated clopidogrel cation, S(+)ClopH$^+$, one picrate anion, Pic$^-$, and a molecule of solvent ethanol *per* asymmetric unit (Figure 2). The cation and anion are linked by a N1-H1...O3 H-bond between the protonated N1 nitrogen of the clopidogrel and the deprotonated phenolate O3 oxygen of the picrate anion [N1...O3 = 2.815(4) Å, N1-H...O3 = 170(4)°, $E_{HB}$ = 2.8 kcal mol$^{-1}$]. This oxygen also acts as an H-bond acceptor O10-H10...O3 from the alcoholic O-H group of the solvent molecule [O10...O3 = 3.047(6) Å, O10-H...O3 = 161°, $E_{HB}$ = 0.6 kcal mol$^{-1}$]. In the crystal structure of Form **1**, weak C-H...O and C-H...Cl H-bonds are also present (Supplementary Table S3). The phenolate group of the picrate anion and the thiophene ring of ClopH$^+$ are almost parallel and form a dihedral angle of 8.6(1)°. The H-N1-C8(chiral)-H fragment displays a *trans* conformation [torsion angle = 159(4)°], as observed in the monoclinic form I of the drug S(+)clopidogrel–hydrogen sulfate and in the S(+)clopidogrel–isopropylsulfate

(Figure 4A,C), but different from the *gauche* conformation present in the other polymorph of clopidogrel hydrogen sulfate used in pharmaceutical formulations, the orthorhombic Form II (Figure 4B). The tetrahydropyridine ring [N1,C1,C2,C3,C4,C5] displays a $^1H_2$ half-chair conformation with the puckering parameters $\varphi_2 = 26.0(5)°$, $\theta_2 = 50.3(4)°$ and $Q_T = 0.521(3)$ Å [41].

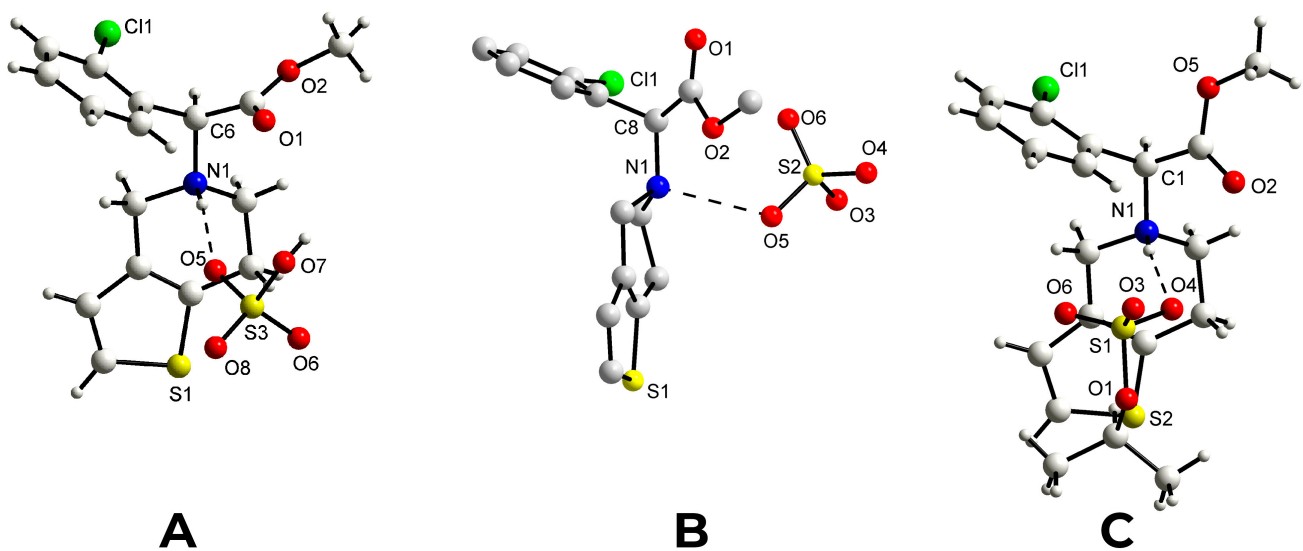

**Figure 4.** Crystal structures of: S(+)ClopH$^+$·HSO$_4^-$ polymorph I (**A**) [6]; S(+)ClopH$^+$·HSO$_4^-$ polymorph II (**B**) [7]; and S(+)ClopH$^+$·Isopropylsulfate (**C**) [10].

Form **2** crystallizes in the *C*2 space group with two independent protonated clopidogrel cations S(+)clopH$^+$ and two picrate anions Pic$^-$ *per* asymmetric unit. The configuration and conformation of the components in both the cation–anion couples are nearly identical (Figure 3); moreover, they are very similar to those found in the crystal of Form **1**.

The N1-H1...O3 H-bond parameters and LS energies are: N1...O3 = 2.752(4) Å, N1−H...O3 = 164(4)°, $E_{HB}$ = 3.4 kcal mol$^{-1}$ in couple A and N1...O3 = 2.770(5) Å, N1-H...O3 = 158(4)°, $E_{HB}$ = 2.7 kcal mol$^{-1}$ in couple B, respectively. In both couples, the phenolate group of the picrate anion and the thiophene ring of ClopH$^+$ are almost parallel, forming dihedral angles of 10.9(1)° and 14.5(1)° in A and B, respectively (Figure 3). In addition, both the H-N1-C8(chiral)-H fragments display the usual *trans* conformation [torsion angles 169(4) and 163(4)°]. The absence of co-crystallized solvent molecules in the crystal lattice favors a more efficient packing, as evidenced by the greater value of the crystal density of Form **2** (1.488 g cm$^{-3}$) with respect to Form **1** (1.457 g cm$^{-3}$) and by non-bonded interactions between cations and anions shorter than the sum of van der Waals radii (Supplementary Table S4). Both the tetrahydropyridine rings show a $^1H_2$ half-chair conformation similar to that observed in the structure of Form **1**, characterized by the following puckering parameters: $\varphi_2(A) = 22.0(6)°$, $\theta_2(A) = 49.7(5)°$ and $Q_T(A) = 0.510(4)$ Å and $\varphi_2(B) = 19.4(6)°$, $\theta_2(B) = 51.3(4)°$ and $Q_T(B) = 0.519(4)$ Å [41].

The calculated X-ray patterns of Forms **1** and **2** were also compared with the corresponding experimental patterns of the obtained samples A-D prepared by slow solvent evaporation and the mechanochemical method (Figure 5). It was found that the XRPD patterns of samples A and C, prepared by solvent evaporation from 98% ethanol and by kneading with 98% ethanol, respectively, corresponded to the theoretical XRD generated from the structural information file of clopidogrel–picrate–monoethanolate (Form **1**). On the other hand, solvent evaporation and mechanosynthesis from *n*-pentanol (sample B and D, respectively) led to co-crystallization toward salts whose structures correspond to Form **2**.

The moderate differences in the positions and intensities of the absorption bands are related to the stretching of the C–H bonds in chlorophenyl rings (3120–3050 cm$^{-1}$)

and in the –OCH$_3$ moiety (2860–2830 cm$^{-1}$), and the C=O stretching (1760–1735 cm$^{-1}$), C–O phenolic stretch (1280 cm$^{-1}$), and the C-N stretching in the clopidogrel fragment (1210 cm$^{-1}$) distinguish the FTIR spectra of salts A and C from those spectra of salts B and D (Figure 6). In addition, the characteristic and strong bands assigned to the FTIR spectra of the clopidogrel free base [42] and the picric acid [43,44] are either shifted or do not appear in the spectra of the A–D samples, confirming complete convergence of the reaction to co-crystallization of the A–D salts.

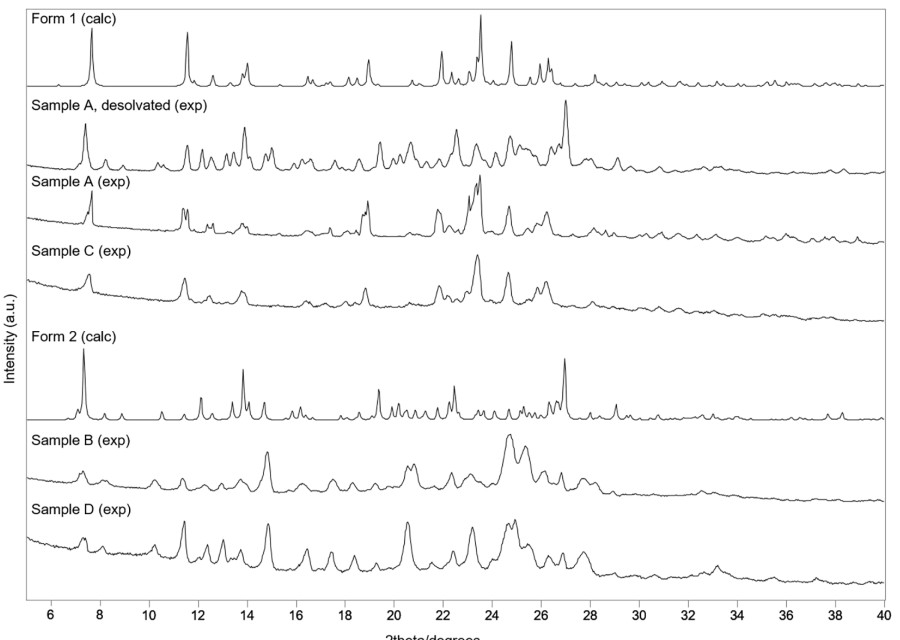

**Figure 5.** Calculated XRPD patterns of Form **1** (determined structure **1** by single-crystal XRD), and Form **2** (determined structure **2** by single-crystal XRD) compared to the experimental XRPD patterns of: Sample A (evaporation in ethanol), Sample C (kneaded with ethanol), Sample B (evaporation in *n*-pentanol), Sample D (kneaded with *n*-pentanol). Sample A was also subjected to DSC analysis up to 120 °C to remove the ethanol molecule and obtain desolvated solid.

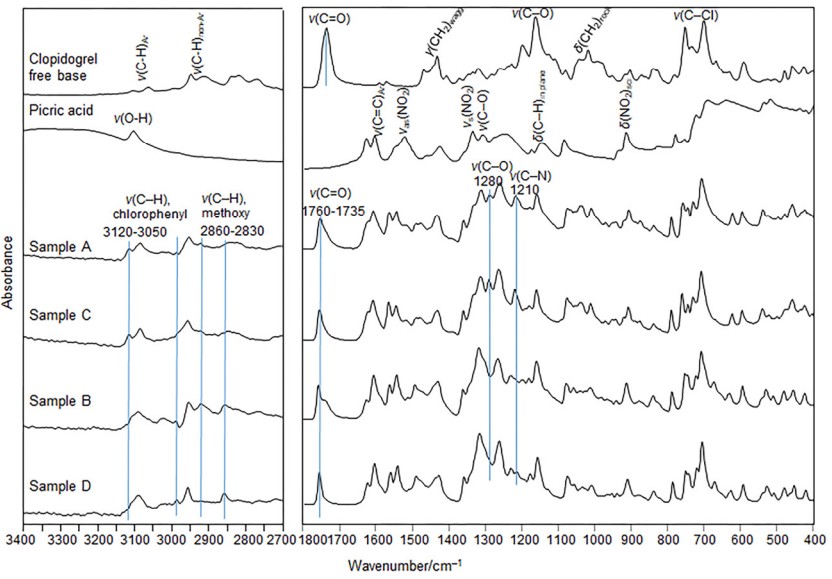

**Figure 6.** FTIR spectra of clopidogrel free base, picric acid, sample A (evaporation in ethanol), sample C (kneaded with ethanol), sample B (evaporation in *n*-pentanol), and sample D (kneaded with *n*-pentanol). The assignment of the main bands and the discrimination bands are marked.

The Raman spectra of samples B and D (Figure 7) are practically identical and indicate the same structure of the clopidogrel–picrate salt prepared by two different methods: evaporation of the solvent and kneading with *n*-pentanol, respectively. The spectral fingerprints showed a shift of characteristic bands present in the pure co-forming substances, clopidogrel free base and the picric acid, whose tentative assignment was provided in agreement with similar molecular structures [45–47].

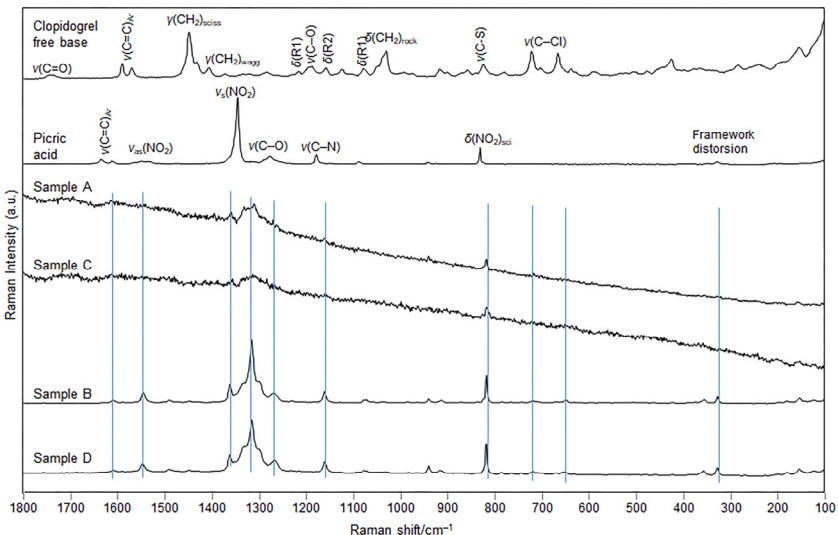

**Figure 7.** Raman spectra of: clopidogrel free base, picric acid, sample A (evaporation in ethanol), sample C (kneaded with ethanol), sample B (evaporation in *n*-pentanol), and sample D (kneaded with *n*-pentanol). The assignment of the main bands and the distinguishing bands are marked.

The most important band shift was noted for the bands most involved in the formation of the salt, $\nu$(C−O), $\nu$(C−N), as well as for the band of the $\nu_s$(NO$_2$). It is worth noting that the Raman spectra also differ from the spectra of the polymorphic Form I and II of clopidogrel–bisulfate [48,49]. On the other hand, the Raman spectra of clopidogrel–picrate–ethanolate (samples A and C), represent similar patterns but exhibit worse spectral features than the spectra of samples B and D. The subtle differences observed between the Raman spectra of the Form **2** samples (B and D) obtained by solvent evaporation and kneading from n-pentanol could be attributed to the obtained smaller particle size and different morphology of the mechanochemically synthesized compound D compared to the solvent-based method. In addition, other factors that governed the spectral changes lie in the kneading procedure, which is a high-energy process, resulting in less crystalline product, evidenced from the somewhat worse Raman spectral baseline of the corresponding product (Figure 7, D sample).

The TG analyses show that the mass loss up to 140 °C is 7.84% for sample A and 8.16% for sample C (Figure 8). These data nicely are consistent with the theoretical mass loss due to the elimination of an ethanol molecule (7.71%) and confirm that samples A and C correspond to structural Form **1** (clopidogrel–picrate–monoethanolate).

The endothermic peak at lower temperature in the DSC curves of the samples A and C indicate the process of evaporation of the ethanol molecule from the compound, while the second endotherm suggests that the melting occurred in a narrow temperature range of 136.5–139.4 °C. It is also evident that the loss of the ethanol molecule begins at lower temperature in sample C compared to sample A, which is depicted in both the DSC and TG curves. This deviation most probably occurs due to a combination of two phenomena: the different particle size of sample A and C (as a result of different sample preparation procedures) and evaporation of small portion of physically adsorbed ethanol on the surface of sample C. On the other hand, the TG curves of samples B and D showed no mass loss (no elimination of the solvate molecule), indicating that these samples adopt the structure

of the non-solvent clopidogrel–picrate salt, Form **2**. The melting endotherm of Sample B occurs at a somewhat lower temperature and with a lower enthalpy than the melting endotherm of Sample D, which can be attributed to the differences in particle size since different procedures were adopted for the sample preparation.

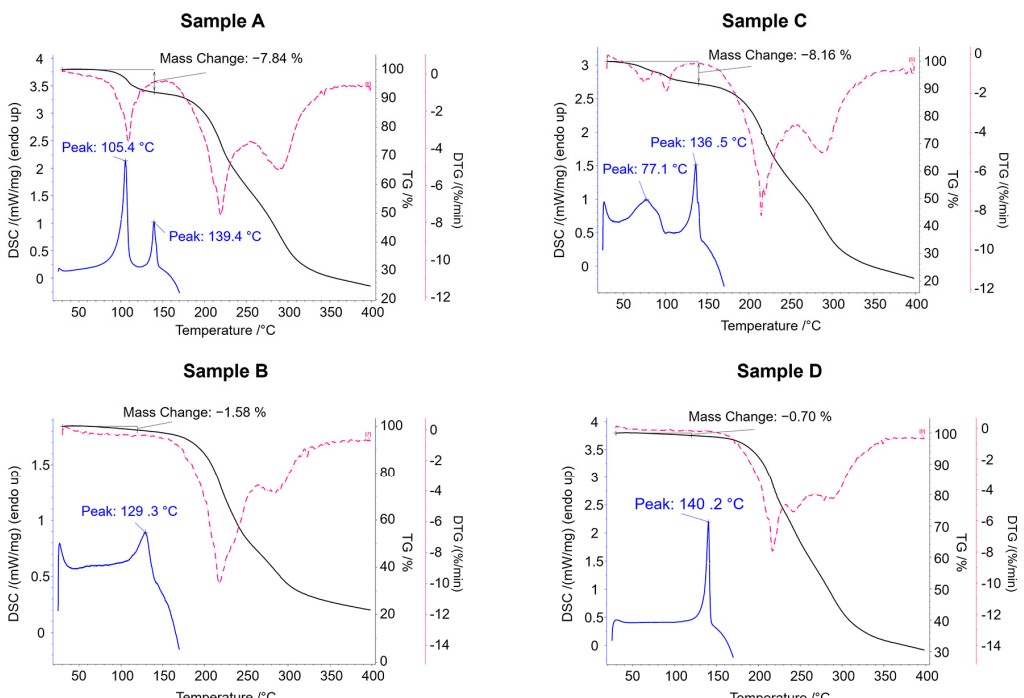

**Figure 8.** TG, DSC, and DTG curves of sample A (evaporation in ethanol), sample C (kneaded with ethanol), sample B (evaporation in *n*-pentanol), and sample D (kneaded with *n*-pentanol).

### *Theoretical Considerations*

From a theoretical viewpoint, in relation to Form **1** (including ethanol molecules in the crystal lattice), its structure was also considered without the presence of solvent molecules. This was in order to obtain further insights into the relevance of incorporating the solvent molecules within the crystal lattice towards overall energetic stabilization, an aspect which could impose certain implications concerning the general tendency of building up solvate crystals. The performed DFTB calculations, however, limited the unit cell parameters to their experimentally determined values to monitor the structural and energetic changes that take place upon exclusion of the solvate molecules from the crystal structure.

Figures 9 and 10 compare the calculated crystal structures of Form **1** and the corresponding non-solvated Form **1**.

The geometry optimization upon elimination of the solvate molecules in the structure indicates substantially different in-crystal molecular arrangements. In the structure of non-solvated Form **1** predicted by DFTB, voids remain at the ethanol positions in the solvate (accompanied by only a slight structural reorganization of the remaining molecular subunits), which means no drastic structure collapse is induced upon elimination of the solvent.

Minor structural adaptations take advantage of the conformational flexibility of the clopidogrelH($^+$) methoxy segment methyl group. Upon removal of ethanol, the closest $H(CH_3) \ldots Cl$ distance increases to 3.39 Å (from 3.04 Å in the case of solvate). Furthermore, the closest O(picrate anion) $\ldots$ S(clopidogrelH($^+$)) contact distance increases from 4.03 Å in the solvate to 4.33 Å in the case of non-solvated Form **1**. The presence of ethanol molecules allows for a more compact structure and higher lattice energy, when compared to the non-solvated Form **1**.

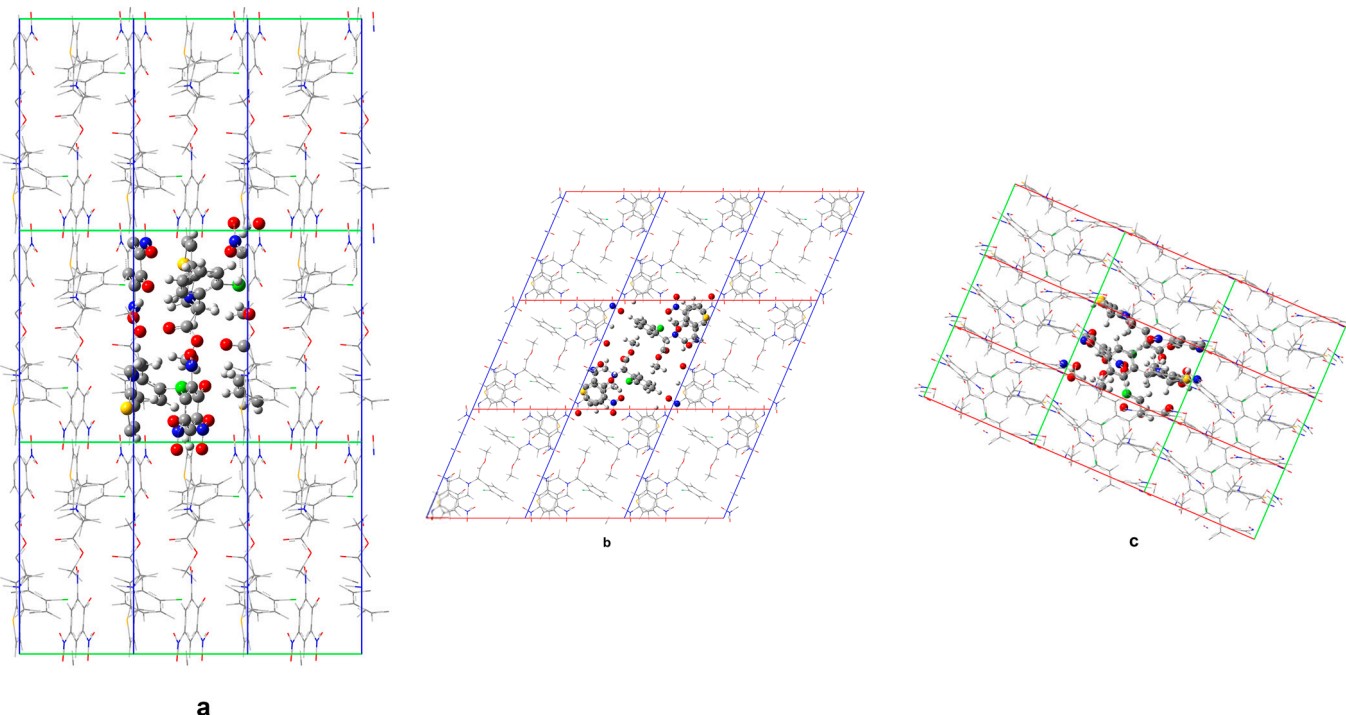

**Figure 9.** The optimized crystal structure of Form **1** with the DFTB3 methodology: (**a**) view along a-axis; (**b**) view along b-axis; (**c**) view along c-axis.

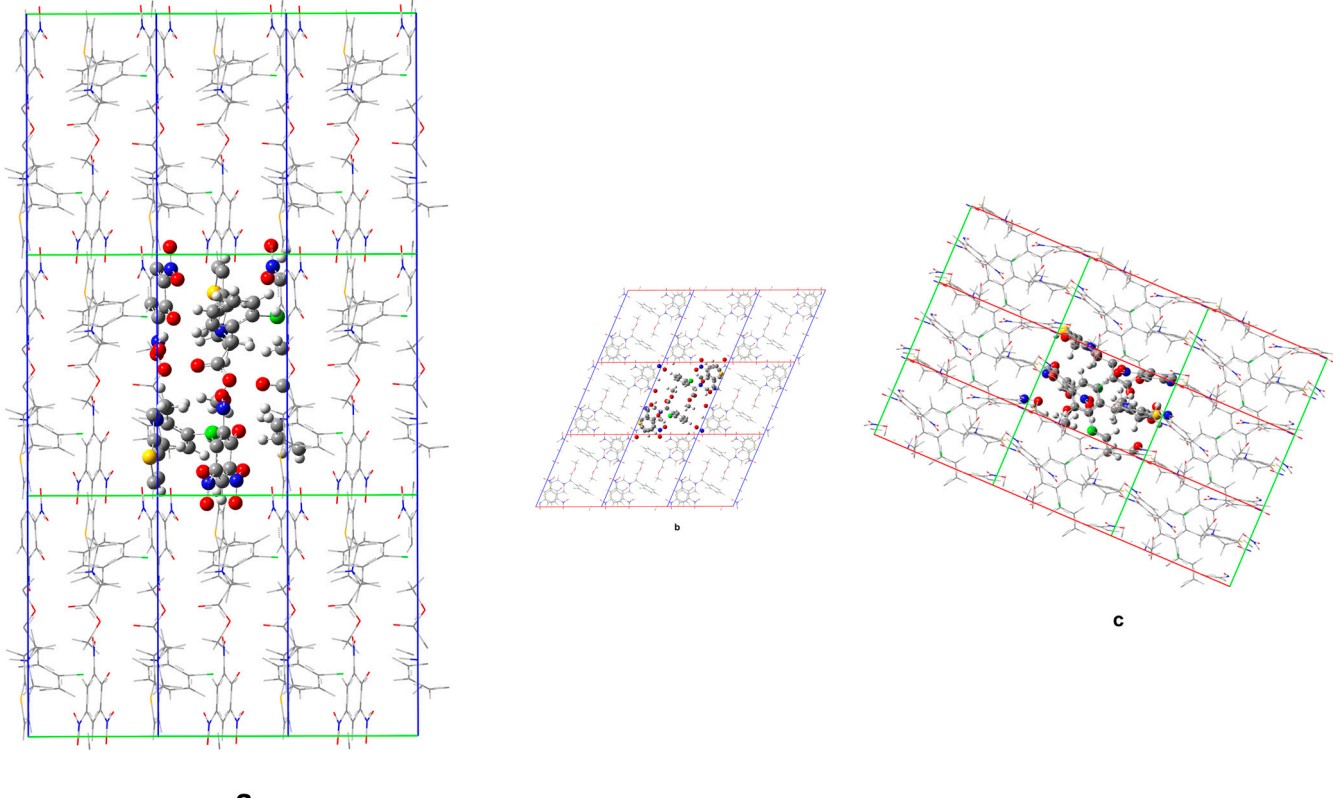

**Figure 10.** The optimized crystal structure of non-solvated Form **1** with the DFTB3 methodology: (**a**) view along a-axis; (**b**) view along b-axis; (**c**) view along c-axis.

The lattice energies in all studied cases were calculated by the following expression (technical details related to the DFTB3 calculations are in Supplementary Material:

$$\Delta E_{\text{latt.}} = \frac{E_{u.c.}}{Z} - E_f$$

where $E_{u.c.}$ denotes the unit-cell energy and $Z$ is the number of formula units per unit cell, while $E_f$ is the total energy of unit cell constituents in the gas phase, isolated from each other (i.e., the sum of energies of unit cell constituents in gas phase, isolated from each other). From a fundamental QM viewpoint, the lattice energy in periodic QM calculations is the expectation value of the crystal Hamiltonian. In essence, it represents the overall stabilization upon formation of the solid phase. Table S5 summarizes the computed lattice energies of Form **2**, Form **1**, and their non-solvated structures. The energy difference between the non-solvated Form **1** and Form **2** is rather small (~3.5 kcal mol$^{-1}$ in favor of Form 2), an order of magnitude, which is quite common for organic polymorph phases [50]. The ranking of the non-solvated Form **1** and Form **2** on the basis of total normalized electronic energy criterion maintains the ordering implied by the lattice energy calculations. Form **2** is energetically favored by ~5.0 kcal mol$^{-1}$. The total energy criterion has often been used as a first step in the stability ranking of polymorphs [50–53]. The included solvent molecules within the crystal of non-solvated Form **1** (the "real" structure of Form **1**), however, increase the lattice energy by about 10.5 kcal mol$^{-1}$. Therefore, the $\Delta E_{\text{latt.}}$ for Form **1** is lower than the corresponding value for Form **2** by ~7.3 kcal mol$^{-1}$. Such trends are in line with the experimentally observed formation of Form **1** instead of the corresponding non-solvated phase.

## 4. Conclusions

Two new pseudopolymorphic forms of S(+)clopidogrel–picrate salt were successfully synthesized and their crystal and molecular structures were determined by single-crystal X-ray diffraction. The asymmetric unit of Form **1** consists of one discrete S(+)ClopH$^+$·Pic$^-$ ionic couple and a molecule of the solvent ethanol, whereas that of Form **2** contains two independent S(+)ClopH$^+$·Pic$^-$ ionic couples. Despite these differences, the geometry of the three ionic couples is almost identical. In particular, the H−N1−C8(chiral)−H fragments of ClopH$^+$ display the same *trans* conformation previously observed in the structures of S(+)clopidogrel–hydrogen sulfate (form I) and S(+)clopidogrel–isopropyl sulfate. In addition, all tetrahydropyridine rings were found in $^1H_2$ half-chair conformation. The phenolate groups of the picrate anions and the thiophene rings of ClopH$^+$ lay on almost parallel planes. Because of the large p$K_a$ difference between picric acid (p$K_a$ = 0.36) [54] and clopidogrel (p$K_a$ = 4.62) [14], $\Delta$p$K_a$ = p$K_a$(D−H) − p$K_a$(A−H$^+$) = −4.26, the outcome of both co-crystallization experiments is a salt, built up by ClopH$^+$·Pic$^-$ H-bonded ionic couples. The reported structures of the pseudopolymorphic Forms 1 and 2 (N···O contact distances in the range 2.752–2.815 Å with LS-computed H-bond energy from 2.68 to 3.45 kcal mol$^{-1}$ and $\Delta$p$K_a$ of −4.26) fit well with the reported correlations between $d$(D···A), $E_{\text{HB}}$, and $\Delta$p$K_a$ [12] in verifying the validity of the p$K_a$ equalization principle. Distinct vibrational spectra (FTIR and Raman), XRPD patterns, and thermal profiles confirmed the structure of Forms **1** and **2** for two pseudopolymorphic salts that correspond to samples A and C as well as to samples B and D, respectively.

According to DFTB3 calculations, the energy difference between Form **2** and the non-solvated Form **1** is predicted to be rather small (~3.5 kcal mol$^{-1}$ in favor of Form **2**). An order of magnitude of this quantity falls within the common range for organic polymorph phases. The total normalized electronic energy criterion leads to the same ranking of Form **2** and the non-solvated Form **1** (Form **2** is again energetically favored by ~5.0 kcal mol$^{-1}$). The DFTB3 level of theory predicts, however, that the included solvent molecules in the Form **1** ("real" Form **1** structure) cause a notable increase in the lattice energy of about 10.5 kcal mol$^{-1}$. This leads to a lower $\Delta E_{\text{latt.}}$ for Form **1** than the corresponding value for

Form 2 by ~7.3 kcal mol$^{-1}$. These predicted trends are in line with the experimentally observed formation of the polymorph **1** solvate instead of the corresponding pure phase.

**Supplementary Materials:** The following supporting information can be downloaded at: https://www.mdpi.com/article/10.3390/cryst14010010/s1, Table S1: Crystallographic data; Table S2: Selected bond distances (Å), bond angles and torsion angles (degrees); Table S3: Hydrogen bonds parameters (Å and degrees); Table S4. Short contacts with distances shorter than the sum of van der Waals radii in Form 2; Table S5: The computed DFTB lattice energies (DElatt.) of Form 1, Form 2, and the simulated non-solvated Form 1. S1: Technical details related to the DFTB3 calculations and CIF files (Polymorph_1.cif and Polymorph_2.cif) have been provided in the supplementary materials.

**Author Contributions:** Conceptualization, A.C. and P.G.; Methodology, A.C.; Software, L.P. and V.B.; Validation, L.P. and V.B.; Formal analysis, A.C., P.M., L.P., M.S.P. and V.B.; Investigation, A.C.; Resources, A.C., P.M., M.S.P. and P.G.; Data curation, A.C., P.M., L.P., M.S.P., V.B., P.G. and L.R.M.; Writing—original draft, A.C., L.P. and P.G.; Writing—review & editing, P.M., P.G. and L.R.M.; Visualization, L.P. and V.B.; Supervision, P.G. and L.R.M. All authors have read and agreed to the published version of the manuscript.

**Funding:** This research received no external funding.

**Data Availability Statement:** The data presented in this study are available in the article.

**Conflicts of Interest:** The author Monika Stojanovska Pecova was employed by the company Alkaloid AD. The remaining authors declare that the research was conducted in the absence of any commercial or financial relationships that could be construed as potential conflicts of interest.

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
