# Peer review of "Polymorphism in S(+)Clopidogrel-Picrate: Insights from X-ray Diffraction, Vibrational Spectroscopy, Thermal Analysis, and Quantum Chemistry"

_crystals, doi:10.3390/cryst14010010_

Round 1

Reviewer 1 Report

Comments and Suggestions for Authors

The manuscript «Polymorphism in S(+)Clopidogrel-Picrate: Insights from X-ray Diffraction, Vibrational Spectroscopy, Thermal Analysis and Quantum Chemistry», by A. Cvetkovski is a very interesting work. Over the past two decades, multicomponent crystals have evolved from a crystallographic phenomenon to a powerful practical tool for improving the bioavailability of difficult-to-solubilise drugs. This paper proves it.

         The work has been carefully done and well structured. I think that the work is of interest for the readership of Crystals and I would be happy to recommend its publication. I have a number of suggestions that might improve the readability of the MS before publication.

Major points:

1.     Two methods in particular stand out in the study of H-bonds - the quantum theory of atoms in molecules (AIM) method of Bader [Bader R (1985) Acc Chem Res 18(1):9] and the natural bond orbital (NBO) analysis of Weinhold [Glendening ED, Landis CR, Weinhold F (2013) J Comput Chem 34(16):1429]. There is no mention of either of these methods in the article.

2.     To accurately assess the energy of a lattice energy basis set superposition error (BSSE) and zero point energy (ZPE) corrections should be applied.

3.     In some co-crystals, a major contribution to the stabilisation of the structure is made by weak non-directional interactions, unfortunately, the authors do not separate the N-H…O and C-H…O contributions to the lattice energy.

4.     What is the role of short contacts (listed in Table 4S) in crystal packing? And how did the authors prove that they are non-bonded?

5.     The authors propose the use of the Lippincott and Schroeder (LS) method (1955) in the article, but why do they not mention alternative methods such as spectroscopic methods, QTAIM or NCI?

6.     Please clarify how Ef is calculated (line 351).

7.     The conclusion is so obvious (p. 11, lines 336-338: The geometry optimization upon elimination of the solvate molecules in the structure indicates substantially different in crystal molecular arrangements). The calculation of the structure without the presence of the solvent molecules seems unnecessary.

Minor points:

1.     Check the spelling of FORM and FROM throughout the text (lines 217, 259)

2.     The energy convergence is too high (10-8 should be used to optimise the crystal and 10-9 should be used for vibrational analysis).

3.     I propose to give the energy of interactions to 1 decimal (lines 221, 223, 249, 250).

4.     Table 5S: there is no comment on how the data in column 2 differ from the data in column 3.

My conclusion is that this article represents a promising discovery, but it would benefit from a major revision. Several of the comments I have made need to be addressed, either by clarifying or correcting misunderstandings, or by making revisions where necessary. Although the article is well written and easy to read, it lacks the depth necessary to fully understand the significance of the findings.

Comments on the Quality of English Language

Minor editing of English language required

Reviewer 2 Report

Comments and Suggestions for Authors

Reviewer's Comments:

1. The URL to access the deposited CIFs should be updated to https://www.ccdc.cam.ac.uk/structures/.

2. The reason for the more pronounced features in Raman spectra, specifically in Figure 7 for Sample B and D, should be further elaborated. The influence of sample composition, crystal quality, and differences in crystalline packing on the inelastic scattering of incident visible monochromatic electromagnetic radiation with a crystal lattice in Raman spectroscopy should be investigated. Exploring these factors could offer insights into the observed differences.

3. The discussion on thermal analysis should include an explanation for the observed slope before the loss of ethanol in Sample C, as shown in Figure 8. Additionally, the authors should address the smaller endothermic peak in the DSC graph for Sample B. Possible explanations for these observations may include differences in sample preparation, the presence of phase impurities, or variations in crystal packing.

4. The subsection title "3.1. Insights from theory" appears abrupt and could benefit from improvement.

5. The authors should clarify the significance of the discussion on the hypothetical non-solvated Form 1 cocrystal with the same unit cell parameters as solvated Form 1. They should explain why it is meaningful to compare the lattice energy calculated from this hypothetical lattice with Form 1 and Form 2. Providing a rationale for considering this hypothetical form and discussing its implications would enhance understanding of its relevance to the study. If these details cannot be provided, it may be advisable to exclude the discussion of this value from the manuscript.

Reviewer 3 Report

Comments and Suggestions for Authors

The manuscript authored by A. Cvetkovski and co-workers presents an intriguing study. The authors synthesized co-crystals of (S)-(+)-clopidogrel and picric acid through various methods. While the commercial drug, Plavix, is its hydrogen sulfate salt, the authors reported the picric acid salt in this manuscript for the first time. Single crystal X-ray analyses revealed two structures: one includes the solvent ethanol in the crystal, and the other is without solvation. The authors evaluated the synthesized solids using various techniques, including Powder XRD, TF-IR, DSC, and more. Additionally, computational calculations highlighted differences between the two polymorphs. I recommend its acceptance for publication in Crystals, but suggest considering some minor aspects to improve the quality of the manuscript.

(1) [Line 93] The free base of (S)-(+)-clopidogrel was acquired by washing the commercial source with aqueous NaHCO3. To assess the purity of this free base, it is recommended to conduct a 1H NMR measurement, along with a specific optical rotation measurement, which can provide an evaluation of the optical purity. In the manuscript, both the measured and literature values of specific optical rotation should be included, along with appropriate citations. This additional information will enhance the characterization of the obtained (S)-(+)-clopidogrel free base.

(2) [Line 235] It may be beneficial to clarify that the crystal structure presentation omits the ethanol molecule for better clarity.

(3) [Figure 8] Based on the TG, DSC, and DTG analyses of ethanolic crystals (Form 1), melting occurred upon the removal of ethanol. The removal of ethanol can be achieved easily by warming the crystal under reduced pressure. In addition to theoretical analysis through computational calculations, direct analysis, such as Powder XRD of the desolvated solid, may be feasible. This could further enhance the depth and completeness of the present manuscript.

(4) It seems that "form 1/2" is incorrectly written as "from 1/2" in several places, so please correct it.

Round 2

Reviewer 1 Report

Comments and Suggestions for Authors

I recommend this article for publication in Crystals